# Discovery of New Secondary Metabolites from Marine Bacteria *Hahella* Based on an Omics Strategy

**DOI:** 10.3390/md20040269

**Published:** 2022-04-18

**Authors:** Shufen He, Peishan Li, Jingxuan Wang, Yanzhu Zhang, Hongmei Lu, Liufei Shi, Tao Huang, Weiyan Zhang, Lijian Ding, Shan He, Liwei Liu

**Affiliations:** 1Li Dak Sum Marine Biopharmaceutical Research Center, Department of Marine Pharmacy, College of Food and Pharmaceutical Sciences, Ningbo University, Ningbo 315832, China; 2011085032@nbu.edu.cn (S.H.); 196000818@nbu.edu.cn (P.L.); 2111085014@nbu.edu.cn (J.W.); 2011085092@nbu.edu.cn (Y.Z.); 2111085060@nbu.edu.cn (H.L.); liufeishi@outlook.com (L.S.); zhangweiyan@nbu.edu.cn (W.Z.); dinglijian@nbu.edu.cn (L.D.); heshan@nbu.edu.cn (S.H.); 2Department of Food Science and Engineering, College of Food and Pharmaceutical Sciences, Ningbo University, Ningbo 315211, China; huangtao@nbu.edu.cn; 3Ningbo Institute of Marine Medicine, Peking University, Ningbo 315800, China

**Keywords:** *Hahella*, prodiginine, chejuenlide, genome mining, LC-MS/MS

## Abstract

*Hahella* is one characteristic genus under the Hahellaceae family and shows a good potential for synthesizing new natural products. In this study, we examined the distribution of the secondary metabolite biosynthetic gene cluster (SMBGC) under *Hahella* with anti-SMASH. The results derived from five genomes released 70 SMBGCs. On average, each strain contains 12 gene clusters, and the most abundant ones (45.7%) are from the family of non-ribosomal peptide synthetase (NRPS) and non-ribosomal peptide synthetase hybrid with polyketide synthase (NRPS/PKS), indicating a great potential to find bioactive compounds. The comparison of SMBGC between *H.* *chejuensis* and other species showed that *H.* *chejuensis* contained two times more gene clusters than *H. ganghwensis*. One strain, designed as NBU794, was isolated from the mangrove soil of Dongzhai Port in Haikou (China) by iChip. The 16S rRNA gene of NBU794 exhibited 99% identity to *H.* *chejuensis* KCTC 2396 and clustered with the *H.* *chejuensis* clade on the phylogenetic trees. Genome mining on strain NBU794 released 17 SMBGCs and two groups of bioactive compounds, which are chejuenolide A-C and nine prodiginines derivatives. The prodiginines derivatives include the well-known lead compound prodigiosin and two new compounds, 2-methyl-3-pentyl-4-O-methyl-prodiginine and 2-methyl-3-octyl-prodiginine, which were identified through fragmentation analysis based on LC-MS/MS. The anti-microbial activity assay showed prodigiosin and 2-methyl-3-heptyl-prodiginine exhibited the best performance in inhibiting *Escherichia coli*, *Salmonella paratyphi* B, MASA *Staphylococcus aureus*, *Bacillus subtilis*, and *Candida albicans*. Moreover, the yield of prodigiosin in *H. chejuensis* NBU794 was also evaluated, which could reach 1.40 g/L under the non-optimized condition and increase to 5.83 g/L in the modified ISP4 medium with macroporous adsorption beads added, indicating that NBU794 is a promising source of prodigiosin.

## 1. Introduction

The ocean covers approximately 71% of the Earth’s surface. As the largest aquatic ecosystem on the earth, the ocean harbors a great number of microorganisms, which occupy more than half of the world’s total prokaryotes [1]. Among the marine microorganisms, bacteria are usually considered as a rich pool for bioactive compounds. The annual report of ‘Marine Natural Products’ reveals that more than 200 new compounds are discovered from marine bacteria each year, and this number reached 300 in 2018, displaying an increasing trend. The natural products produced from marine bacteria include peptides, polyketides, terpenes, alkaloids, etc., which exhibit various bioactivities, such as anti-infection, anti-cancer, and protease inhibition. However, the data show that the majority of new compounds discovered from marine bacteria are from *Streptomyces* and cyanobacteria, which account for 76% in total [2], while a small proportion of natural products are discovered from other bacteria.

*Hahella* is identified by Lee et al. (2001) as a Gram-negative, facultative anaerobic marine bacteria from the family of Hahellaceae. They are mobile by means of one single polar flagellum and only live in slightly halophilic environments (2% NaCl). Up to now, three species in this genus have been identified, including *H. ganghwensis*, *H. antarctica*, and *H. chenjuensis* [3,4,5]. One known study showed that *H. chejuensis* could cause the red egg disease in tilapia (*Oreochromis* spp.) [6], indicating that *Hahella* might be one kind of marine pathogen. When this current study was performed, only 17 *Hahella* 16S rRNA genes and 5 genomes were documented in the NCBI database (https://www.ncbi.nlm.nih.gov/ (accessed on 14 December 2021).), so little is known about this genus.

Even though only a few strains in *Hahella* have been uncovered, they have shown a good potential for synthesizing bioactive compounds. The prodiginine derivatives, including well-known prodigiosin, are the first type of natural product identified in *H. chejuensis* (Appendix A), which are characterized with the core structure of prodiginine including three continuous pyrrole rings [7,8]. As one star molecule in this family, prodigiosin has a wide array of biomedical and industrial applications, including algicidal, antibacterial, anticancer, antimalarial, antiprotozoal, colorants, immunosuppressive agents, and insecticides [9]. Moreover, prodigiosin has been reported to exhibit no damage to chromosomes and no acute and hereditary toxicity in Kunming mice at a dose of 10 g/kg, suggesting a good selectivity on molecular targets and biological safety [10]. Actually, prodigiosin is first identified in terrestrial bacterium *Serratia marcescens* and confirmed in many different marine bacteria later on, such as *Pseudoalteromonas*, *Streptomyces*, and *Hahella* [11], indicating a wide distribution in nature. Up to now, eight prodiginine derivatives including prodigiosin have been identified in *H. chenjuensis* KCTC 2396 [8].

Chejuenolide A-C are 17-membered carbocyclic tetraenes, which are the second type of bioactive compound identified in *H. chejuensis* (Appendix A) [12,13]. Their structures are very similar to the antibiotic lankacidin C produced by *Streptomyces*, except for a δ-lactone ring and a pyruvyl group connected to C-18 via a nitrogen atom (Appendix A). However, the minor structural differences result in totally different bioactivities. In comparison with the antibiotic lankacidin C, chejuenolides exhibit no antibiotic activity, but a strong inhibitory effect on tyrosine phosphatase 1B [12]. The biosynthesis scheme of chejuenolides has been elucidated through heterologous expression and bioinformatics analysis [14]. The gene cluster of chejuenolides consists of six genes, coding one non-ribosomal peptide synthetase (NRPS) module, five polyketide synthase (PKS) modules, and one stand-alone trans-AT domain, which could be assigned to the iterative trans-AT PKS family.

Although two type of bioactive compounds have been discovered, we still have little knowledge of the real resource of secondary metabolites in *Hahella*. We hypothesized that *Hahella* might contain more valuable secondary metabolites than previously shown. Herein, we studied the SMBGC distribution in *Hahella* with genome mining and uncovered the unbalanced distribution of SMBGCs among different species. Furthermore, we isolated one new strain (*H. chenjuensis* NBU794) and discovered nine prodiginine derivatives including two new molecules and chejuenolide A-C through metabolomics.

## 2. Results

### 2.1. SMBGC Analysis in Hahella

Genome mining has become a key strategy to guide the discovery of new natural products [15,16,17,18]. Here, we collected five genomes from the NCBI database, including *H. ganghwensis* DSM 17,046, *H. chejuensis* KCTC 2396, *H. chejuensis* KA22, *H. chejuensis* HN01, and *Hahella* sp.CCB-MM4 (Appendix A). The anti-SMASH analysis identified 70 SMBGCs, and each genome contained 12 SMBGCs on average (Appendix A). It has been shown that secondary metabolites synthesized by NRPS and PKS are usually considered as a good resource for bioactive compounds [19]. Here, we found that 45.7% of SMBGCs in *Hahella* belong to the family of NRPS and NRPS/PKS hybrid (Appendix A).

In total, 15 NRPS gene clusters were identified in this study and classified into eight groups based on the gene composition. Only the group I-III contained a single NRPS module, while all the others possess continuous NRPS modules, showing a good potential to find peptidic compounds with complicated structures (Figure 1A). One epimerization domain(E), which could convert the configuration of amino acid from L to D, was identified in the gene cluster of group VII (Figure 1A), indicating that the final product might contain D-amino acid. In addition, 17 NRPS/PKS hybrid gene clusters were identified and assigned to six groups based on the gene composition (Figure 1B). The group I contained one single PKS module flanked by an adenylation domain and a thiolation domain, while the others contained multiple PKS and NRPS modules, such as the groups IV, V, and VI. Moreover, a variety of modification domains, for example, KR (ketoreductase), DH (dehydratase), ECH (enoyl-CoA hydratase/isomerase), ER (enoyl-reductase), and cMT (carbon methyltransferase) (Figure 1B), were detected in the NRPS/PKS hybrid gene clusters, suggesting a great diversity in the final structures. Moreover, we also found some other types of SMBGCs, such as ectoine, NAGGN (N-acetylglutaminylglutamine amide), betalactone, PBDE (Polybrominated diphenyl ether cluster), hserlactone, CDPS (tRNA-dependent cyclodipeptide synthases), butyrolactone, prodigiosin, siderophore, and RiPPs (Ribosomally synthesized and post-translationally modified peptides) (Appendix A), indicating a great chemical diversity in *Hahella*.

Even though *Hahella* consists of three species, only *H. chejuensis* and *H. ganghwensis* have genome information available in the NCBI database. Therefore, we undertook a comparison of SMBGCs between *H. chejuensis* and *H. ganghwensis* and found that the SMBGCs in *H. chejuensis* were almost two times greater than *H. ganghwensis* (Appendix A).

### 2.2. New Strain Isolation and Identification

By considering the high number of SMBGCs in the genome, we selected *H. chejuensis* as our target to find new bioactive compounds. The in situ culture technique (Ichip) combined with PCR screening was applied to search for new strains from mangrove soil samples collected from Dongzhai Port in Haikou, Hainan Province, China. Finally, one new strain named as NBU794 was found. It exhibited a dark red color on both the agar plate and liquid culture. We sequenced the 16S rRNA gene of NBU794 and blasted it against the NCBI database. The sequence data exhibited 99.6% similarity between NBU794 and *H. chejuensis* KCTC 2396. To study the taxonomic status of NBU794, we undertook a phylogenetic analysis of the 16S rRNA tree of NBU794 and another 17 strains in two modes, which were neighbor-joining and maximum likelihood (Figure 2 and Appendix A). On the trees, NBU794 apparently clustered with *H. chejuensis* and was separated from *H. ganghwensis*. The cell morphology of NBU794 was examined using transmission electron microscopy. We found that NBU794 grown in M9 and 2216E medium had no flagellum and cells were long rods, which was similar to type strain *H. chejuensis* KCTC 2396 (Figure 2). The ANI value between NBU794 and type strain *H. chejuensis* KCTC 2396 was investigated, and the 88.2% value was below the thresholds (90%) recommended for species delineation [20]. Therefore, we assigned NBU794 as a new strain of *H. chejuensis* initially according to the phylogenetic analysis and morphology description.

### 2.3. Genome Sequencing and SMBGCs Prediction in NBU794

The whole genome sequencing and assembly of *H. chejuensis* NBU794 were performed by using PacBio RSII/Sequel SMRT instrument and Illumina HiSeq 4000 platform at BGI company (Shenzhen, China). The full sequence data was uploaded to the web version of anti-SMASH 6.0 for the prediction of SMBGCs, and 17 SMBGCs were identified (Appendix A). Among the 17 detected SMBGCs, only four exhibited more than 50% similarity to the known biosynthetic gene clusters in the Minimum Information about a Biosynthetic Gene cluster database [21], while the others exhibited less than 20% similarity, indicating a good chance to find new natural products (Appendix A).

### 2.4. Prodiginines Identification

To find new natural products in the NBU794 strain, we applied the one strain many compounds (OSMAC) strategy [22]. The NBU794 strain was grown in different mediums, including M9, R2A, ISP4, and 2216E. The fermentation extracts were analyzed with LC-MS/MS-based metabolomics method. Interestingly, one prominent UV Peak (designed as Peak #7) was detected at retention time of 24 min (Figure 3A). The λmax of the UV-Vis spectrum was 534 nm. The LC-MS analysis released a molecular ion of 324.2067 [M + H] ^+^ corresponding to a molecule formula C_20_H_25_N_3_O. The Global Natural Products Social Molecular Networking (GNPS) analysis showed that the compound matched with the red pigment antibiotic, prodigiosin (Figure 3B and Appendix A). To confirm the prediction, we compared it with standard prodigiosin in *H. chejuensis* KCTC 2396, and found they shared the same retention time, UV-Vis spectrum, and high resolution mass spectrum data (Appendix A). Coincidently, the biosynthetic gene cluster of prodigiosin was also identified in the NBU794 strain. Therefore, based on the genome data (Appendix A) and chemical analysis (Figure 3), we confirmed that the studied compound was prodigiosin.

In addition to prodigiosin, we detected another eight UV peaks sharing similar UV-Vis absorption spectra, but displayed different molecular ions (Figure 3 and Appendix A). The GNPS analysis showed that five ions, including prodigiosin, connected to each other on the network, while the other three ions existed independently (Appendix A). Therefore, we suspected that they were prodiginine derivatives. To confirm our prediction, we introduced the targeting LC-MS/MS experiment, which is a commonly used method to identify the structures of prodiginine derivatives. The five compounds in Peaks #1, #4, #5, #8, #9 released five molecular ions of 296.1759(C_18_H_21_N_3_O), 338.2216(C_21_H_27_N_3_O), 366.2178(C_23_H_31_N_3_O), 310.1909(C_19_H_23_N_3_O), and 352.2380(C_22_H_29_N_3_O) in the high resolution LC-MS (Appendix A), and their fragmentation patterns were similar to prodigiosin (Figure 3B). The mass differences could be attributed to the different alkyl chains, such as propyl, butyl, hexyl, heptyl, and octyl side chains. Through the comparison of mass spectra data and UV-Vis absorption spectrum with the previous reports [8,23], we confirmed five compounds (Peak #1, #4, #5, #8, #9) as 2-methyl-3-propyl-prodiginine, 2-methyl-3-hexyl-prodiginine, 2-methyl-3-octyl-prodiginine, 2-methyl-3-butyl-prodiginine, and 2-methyl-3-heptyl-prodiginine(Figure 3B). The compound in Peak #6 released a molecular ion of 394.1896[M + H] ^+^. Its resulting ion scan mass spectrum contained a *m/z* 238.0859[M-156], 252.1000[M-142], 336.1476[M-58], 362.3031[M-32], which was consistent with previous data derived from undecylprodiginine [8] (Figure 3B).

We also detected two new compounds in Peaks #2 and #3, which released two molecular ions 340.2015[M + H] ^+^ and 354.2171[M + H] ^+^, respectively (Figure 4). Interestingly, they exhibited a similar UV spectrum to known prodiginine derivatives. To elucidate their structures, we tried to isolate two compounds to run an NMR experiment, but their amount in the fermentation extract was too low, and the compounds were so sensitive to light that they kept degrading all the time, which prevented us from obtaining clean NMR data. Therefore, we introduced the LC-MS/MS method, which was a well-established and frequently used strategy for studying prodiginine derivatives. In comparison with prodigiosin (324.2067, [M + H] ^+^), we predicted compound in Peak #2 to contain one more oxygen based on the molecular weight difference. To determine the location of one extra oxygen atom, we checked the MS/MS data and found the fragment 282 (Figure 4), indicating a loss of butyl alkyl chain from the parent molecule. On the contrary, prodigiosin contained a fragment of 252, suggesting a loss of pentyl alkyl chain. The 30 dalton difference between fragments 282 and 252 indicated that the new compound might contain one second O-methyl group. To our surprise, we found two characteristic fragments of 161.0702 and 175.3624, which excluded all the possibilities except for one position, 3-C on the Ring C (Figure 4). Therefore, we identified the new compound in Peak #2 as 2-methyl-3-propyl-4-O-methyl-prodiginine. Because of the 14 dalton difference between compounds in Peak #2 and Peak #3, when we replaced the butyl side chain with the pentyl side chain, the MS/MS data of the compound in Peak #3 matched with the structure of 2-methyl-3-pentyl-4-O-methyl-prodiginine.

### 2.5. Anti-Microbial Assay

Naturally occurring prodiginines are a large family of secondary metabolites, which can inhibit many different microorganisms. In this study, we grew *H. chejuensis* NBU794 in 70L M9 medium and isolated four pure prodiginine derivatives, including prodigiosin, 2-methyl-3-propyl-prodiginine, 2-methyl-3-butyl-prodiginine, and 2-methyl-3-heptyl-prodiginine. In the agar diffusion assay, we chose *S. paratyphi* B, *P. aeruginosa*, Methicillin-resistant *S. aureus*, *B. subtilis*, *E. coli*, and *C. albicans* as the indicative strains. The results showed that all four compounds could inhibit Methicillin-resistant *S. aureus* and *B. Subtilis* but failed to kill *P. aeruginosa* (Figure 5). To our surprise, the prodigiosin and 2-methyl-3-heptyl-prodiginine also gave a clear inhibition zone on the agar plate with *E. coli*, *S. paratyphi* B, and *C. albicans* (Figure 5).

Moreover, we also tested the IC_50_ values of isolated prodiginine derivatives. The data was included in Table 1. Both prodigiosin and 2-methyl-3-heptyl-prodiginine had the best performance against all indicative strains except *P. aeruginosa*, and their IC_50_ ranged from 1.56 to 50 µg/mL. Regarding the other two compounds, they had a relatively higher IC_50_ against *B. subtilis* (12.5–25 µg/mL) and Methicillin-resistant *S. aureus* (50 µg/mL), but failed to inhibit *E. coli*, *S. paratyphi* B, and *C. albicans* (Table 1).

### 2.6. Production of Prodigiosin

Prodigiosin has been proved to possess various bioactivities, such as anti-bacteria, anti-cancer, algicide, and so on, showing a great potential to be developed into a lead compound. Therefore, it is very necessary to find a strain with a high yield of prodigiosin. Since the prodigiosin was the major component in *H. chejuensis* NBU794, we grew the strain under different conditions and found the production of prodigiosin in ISP4 medium was the best, which could reach 1.40 mg/mL without any optimization (Appendix A). When we added HP20 (Macroporous Adsorption Resin) in ISP4 and grew the strain at the same growth condition, the prodigiosin production increased to 3.43 g/L (Appendix A). Since the different carbon sources have been proven to affect the yield of prodigiosin [24], we then introduced glucose and sucrose as the carbon source to replace starch in ISP4 and the yield of prodigiosin in *H. chejuensis* NBU794 further increased to 3.46 and 5.83 g/L, respectively (Appendix A).

### 2.7. Discovery of Chejuenolide A–C

In addition to prodiginines, we also found another three new characteristic UV peaks at a retention time of 15–18 min compared with the blank control (Appendix A). Their λmax absorption was 210 nm, and released three molecular ions, 388.2483 (Peak A), 388.2489 (Peak B), and 388.2459 (Peak C), which corresponded to a predicted formula C_23_H_33_NO_4_. In order to elucidate their structures, we isolated the compound in Peak A and ran it with NMR. The ^1^H-NMR and ^13^C-NMR data were completely the same as chejuenolid A (Appendix A). We further compared fermented extracts of *H. chejuensis* NBU794 with *H. chejuensis* KCTC 2396, which has been proved to produce chejuenolide A–C, and found the other two compounds in Peaks B and C had the same retention time as the chejuenolide B and C in *H. chejuensis* KCTC 2396 (Appendix A). Therefore, we confirmed that the three discovered compounds were chejuenolide A–C.

## 3. Discussion

As a good resource of natural products, marine bacteria have received a lot of attention. Even though hundreds of new compounds from marine bacteria are identified each year, most of them are from a few well-known producers, such as *Streptomyces* and cyanobacteria [2]. A previous study shows that 99% of microorganisms in nature are still uncultured [25]. Therefore, discovering natural products from marine uncultured bacteria could be a new option [26].

*Hahella* was first identified at the beginning of the 21st century. Only three new species have been identified up to now, so most of the members under this genus are still unknown. During the last two decades, several studies were performed to investigate *H. chejuensis* and a number of new bioactive compounds were found, which developed the research interests [7,8,12,13]. In this current study, we investigated *Hahella* with genome mining and found that *Hahella* contained 12 SMBGCs in each genome on average. Even though the SMBGC number in *Hahella* is not as high as the *streptomyces* and cyanobacteria, it is much higher than the average number (5) derived from 200,000 microbial genomes [27], indicating that *Hahella* is a good resource for new natural products. However, the comparison of SMBGC numbers between two different *Hahella* species showed that SMBGC in *H. chejuensis* was twice as high as in *H. ganghwensis*. The phylogenetic tress clearly showed that both species formed two separated clades, indicating they might have different evolution paths. Even though we have no idea why two *Hahella* species have a great difference in terms of SMBGCs distribution, it is widely accepted that the microorganism that contains rich secondary metabolites might have more advantages in natural selection, while the one with fewer secondary metabolites is more prone to acquiring nutrients or protection by forming a close connection with other species or a host, because this is more economical [28,29,30,31].

In order to find new natural products from *Hahella*, we applied genome mining and metabolomics to analyze a new isolated strain NBU794. Finally, two new compounds and seven known prodiginine derivatives were identified. The prodiginines are one group of microbial secondary metabolites containing one tripyrrole core structure (Appendix A) [32]. They not only exhibit good bioactivities, but also diverse structures through the modification on the third pyrrole ring, such as methylation, alkylation, and cyclization [8,33]. Our study uncovered two new compounds containing a second O-methyl group on the C-3 of the C-ring (Appendix A), indicating the existence of a new structure diversification mechanism. Kim et al. (2007) found that dipyrrolyldipyrromethene prodigiosin contains a second O-methyl group on the third pyrrole ring, but dipyrrolyldipyrromethene prodigiosin had a fourth pyrrole ring instead of one alkyl side chain (Appendix A), indicating a different biosynthesis pathway [8]. In order to provide a deep understanding of how new compounds were biosynthesized, we compared the prodiginines gene cluster within NBU794 to the homologous one in *H. chejuensis* KCTC 2396 (Appendix A). However, the almost 100% similarity suggested that prodiginine biosynthesis enzymes might have a broad substrate specificity, which could result in new minor compounds during the biosynthesis.

In the present study, we tested four prodiginine derivatives and found that they showed a good performance against Gram positive bacteria, such as MRSA *S. aureus*, *S. paratyphi* B, and *B. subtilus*. Moreover, both prodigiosin and 2-methyl-3-heptyl-prodiginine exhibited a clear inhibition on *E. coli* with a low IC_50_, suggesting that they could be developed into a wide-spectrum antibiotics in the future. In contrast, the other two prodiginine derivatives (2-methyl-3-propyl-prodiginine and 2-methyl-3-butyl-prodiginine) were not effective in the anti-microbial test. Based on the anti-microbial activity and their structures, we speculated the alkyl chain is the key functional group that affects prodiginines’ anti-microbial activities, and the prodiginine with a pentyl group gave the best performance in the test (Appendix A and Figure 4). The alkyl chain has been proven to be compulsory for anti-bacteria activity of many antibiotics, such as surfactins, bacillomycin L, and Pseudodesmin A [34,35,36]. The alkyl chain with a different length has a strong effect on the antibiotic activity. Therefore, our results not only support the previous study, but also provide a guideline for the optimization of structures and activities on prodiginine derivatives.

Considering the good bioactiviy of prodigiosin, there is a great potential to develop prodigiosin into one drug lead. Therefore, it is very important to find a strain with a great yield of prodigiosin. A recent study suggested that the yield of prodigiosin in *H. chejuensis* could be increased to 2.5 g/L by using glucose as the carbon source [24]. However, our results showed that the prodigiosin in *H. chejuensis* NBU794 could reach 5.83 g/L in a modified ISP4 medium with HP20 beads (Appendix A). The HP20 beads belong to highly porous synthetic adsorbent resin. In the fermentation, HP20 could reduce the concentration of secondary metabolites in the culture due to strong binding in order to avoid feedback inhibition and promote a high production of the target molecule. Our data not only showed a high yield of prodigiosin in *H. chejuensis* NBU794, but also provided a good way to increase prodigiosin production.

Although we made some important discoveries in this study, there are still some limitations. First, the genome mining data in this study were derived from a small number of genomes. It will be more reliable to make a conclusion with more data. Therefore, more *Hahella* stains need to be studied and sequenced in the future to help us understand the real resource in the *Hahella* genus and guide the discovery of new compounds. Second, we found nine prodiginines in NBU794 strains, but only prodigiosin was abundant in the crude extract. Because of the limited production, we hardly tested anti-microbial activity of all compounds, including two new compounds, which stops us from understanding their biological functions. In addition, we also met the same problem with chejuenolides in NBU794. There are some minor peaks sharing similar UV-Vis spectra and molecular weights with chejuenolide A-C. Because of the extremely low yield, we hardly ruled out their structure with natural product chemistry techniques. Maybe the large-scale fermentation in the future could resolve the problem. Third, except for prodiginines and chejuenolides, most SMBGCs in the NBU794 strain were still silent. However, we believe that SMBGC silent problems could be bypassed in the future using the fast development of new synthetic biology techniques in *Hahella*.

## 4. Materials and Methods

### 4.1. Anti-SMASH Analysis

The genomes of *H. ganghwensis* DSM 17,046, *H. chejuensis* KCTC 2396, *H. chejuensis*. KA22, *H. chejuensis*. HN01, and *Hahella* sp.CCB-MM4 were downloaded from the NCBI database. The detailed information of five genomes is listed in Appendix A. All genome sequences with Fasta format were uploaded manually to the website version anti-SMASH 6.0 for BGC prediction with default parameter settings [15].

### 4.2. Sampling and Strain Isolation

The strain isolation was carried out with Epstein method [37]. A total of 1 g of mangrove sediment collected from Dongzhai Port in Haikou (China) was weighed and mixed with 10 mL sterilized water. The mixture was vortexed for 10 min to create a soil slurry. The slurry was left to settle for 5–10 min to allow larger particles to settle. The supernatant was diluted with sterilized water by 104 times. Then, 2.5 mL of appropriate dilution was mixed with 22.5 mL of molten ISP4 agar and vortexed for 15 s. The agar-sample mix was transferred into a sterile pipette basin and dispensed with a multichannel pipette into each well of the ichip. When the agar was solid, the ichip was sealed by sterilized polycarbonate membrane with a 0.03 µm pore size and dried in air for 30 min. The ichip devices were cultured in situ in mangrove sediments in Haikou City, Hainan Province, China for 20 days. At the end, the ichip was collected and rinsed with sterile water. After opening the ichip in a Lamina flow hood, a toothpick was used to touch the colony and streak on 50% R2A agar in order to obtain a single colony. The single colonies with great morphological differences were selected for colony PCR analysis with 16S rRNA universal primers. The primer sequences are 27F 5′-AGAGTTTGATCCTGGCTCAG-3′ and 1492R 5′-GGTTACCTTGTTACGACTT-3′ [38]. The PCR products were sequenced and blasted against the Standard Nucleotide BLAST of National Center for Biotechnology Information (NCBI, https://blast.ncbi.nlm.nih.gov/Blast.cgi (accessed on 1 March 2021)).

### 4.3. Strain Culture Condition

The NBU794 strain was grown in M9, R2A, ISP4, and 2216E liquid medium with 2% sea salt, and cultured in a 180 rpm/min shaker at 28 °C for 3–5 days. The agar plates of *S. paratyphi* B, *P. aeruginosa*, *S. aureus*, Methicillin-resistant *S. aureus*, and *B. subtilis* were grown at 37 °C for 18 h. *C. albicans* was grown on a YPD agar plate at 28 °C for 30 h. The ISP4 medium was purchased from ELITE-MEDIA (Shanghai, China), YPD and LB medium were purchased from Hopebiol (Qingdao, China), and the other reagents were purchased from Macklin (Shanghai, China).

### 4.4. Strain Identification

The *H. chejuensis* NBU794 genomic DNA was extracted from three-day culture using the Magen HiPure Bacterial DNA Kit. The genome sequencing and assembly was performed by Shenzhen BGI company in China. The 16S rRNA gene was retrieved from the full genome data and introduced for a BLAST search against sequences in GenBank. The 16S rRNA genes from the NBU794 strain and another 17 16S rRNA gene sequences downloaded from the NCBI database were used to construct a phylogenetic tree by using the software MEGA version 7.0 after multiple alignments using CLUSTAL_W and clustering with the neighbor-joining and maximum likelihood methods [39]. Cell morphology of the NBU794 strain was examined using transmission electron microscopy (TEM, H-7650; Hitachi) after negative staining with saturated uranyl acetate. The ANI value between strain NBU794 and *H. chejuensis* KCTC2396T was calculated by using the ANI calculator online service (https://www.ezbiocloud.net/tools (accessed on 1 March 2021)).

### 4.5. Compound Purification

*H. chejuensis* NBU794 strain was inoculated in 70L M9 medium and cultured for 5 days at 28 °C with a speed of 180 rpm/min. The 70L culture were extracted with equal volume ethyl acetate twice and dried with rotating evaporator. The dried extracts were mixed with 3.25 g silica powder and dried at 70 °C for one hour and subjected to open column chromatography on silica gel (300–400 mesh, column: 7 cm [Length] × 5 cm [Diameter]). A total of five different solvents, including petroleum ether (100%), petroleum ether/ethyl acetate (7:3), petroleum ether/ethyl acetate (1:1), ethyl acetate (100%), and methanol (100%), were used to wash the silica column with three column volumes each time. Finally, we collected 36 fractions and analyzed them with HPLC. The fractions were injected into a Luna C18 column (100 × 3 mm, 2.6 μm, Phenomenex) in 50 μL batches. The column was eluted in a gradient from 10% aqueous MeCN to 100% MeCN in 20 min at 25 °C and washed with 100% MeCN between the injections. Finally, the prodiginines were detected in NO.10–18 fractions. The NO.10–18 fractions were evaporated and suspended in 50% acetonitrile for the further semi-HPLC purification. The prodiginine extracts were injected into a Luna C18 column (250 × 10 mm, 5 μm, Phenomenex) in 400 µL batches. The column was eluted with a gradient from 42% to 60% aqueous MeCN in 55 min at 25 °C and washed with 100% MeCN between the injections. The fractions containing prodiginines were pooled and evaporated. Moreover, the fraction NO.19–27 from silica chromatography was confirmed to contain chejuenolides. We evaporated fraction NO.19–27 and suspended it in 30% MeCN. The chejuenolide samples were injected into a Luna C18 column (250 × 10 mm, 5 μm, Phenomenex) in 400 µL batches. The column was eluted isocratically with 24% aqueous MeCN at 25 °C for 43 min and washed with 100% aqueous MeCN between the injections. The fractions containing chejuenolides were pooled and evaporated. The chemical solvents were HPLC levels and purchased from Fisher Scientific.

### 4.6. LC-MS Analysis

The LC-MS analysis was performed on an Agilent 6545 Quadrupole Time-of-Flight LC/MS. The testing samples were injected into a Luna C18(2) column (100 × 3 mm, 2.6 μm, Phenomenex) and eluted from 10% aqueous MeCN to 100% MeCN in 20 min. The flowing speed of the eluent was 0.7 mL/min with 0.1% formic acid. Mass spectra were acquired using electrospray ionization in the positive mode. Spectra were recorded using a scan range from 200 to 3200 *m/z*, and targeted MS/MS spectra were recorded as averages of three spectra. The precursors were fragmented into product ions with various CEs, including 13 ev at 250 *m/z*, 20 ev at 350 *m/z*, and 27 ev at 450 *m/z*. The DL temperature was 200 °C, the nebulizing gas flow rate was 3 L/min, the heat block temperature was 350 °C, the drying gas flow rate was 15 L/min, the column oven temperature was 30 °C, and the cooler temperature was 5 °C.

### 4.7. Molecular Networking Analysis

The targeted LC-MS/MS data of prodiginine derivatives were converted into mzXML with MSConvert and uploaded to Global Natural Products Social Molecular Networking (GNPS, http://gnps.ucsd.edu (accessed on 4 June 2021).) using File Zilla Software. The molecular networking was established using the online workflow with the default parameter settings. The data was filtered by removing all MS/MS fragment ions within +/−17 Da of the precursor *m/z*. MS/MS spectra were window filtered by choosing only the top 6 fragment ions in the +/−50Da window throughout the spectrum. The precursor ion mass tolerance was set to 2.0 Da and a MS/MS fragment ion tolerance of 0.5 Da. A network was then created where edges were filtered to have a cosine score above 0.7 and more than 6 matched peaks. Further, edges between two nodes were kept in the network if, and only if, each of the nodes appeared in each other’s respective top 10 most similar nodes. Finally, the maximum size of a molecular family was set to 100, and the lowest scoring edges were removed from molecular families until the molecular family size was below this threshold. The spectra in the network were then searched against GNPS spectral libraries. The library spectra were filtered in the same manner as the input data. All matches kept between network spectra and library spectra were required to have a score above 0.7 and at least 6 matched peaks [40].

### 4.8. NMR

NMR spectra of chejuenolide A were collected on a Varian Inova 600 NMR spectrometer and the solvent was CD_3_OD.

### 4.9. Antimicrobial Activity

The anti-microbial activity of prodiginine derivatives was determined by the agar diffusion assay and MIC with a modification [41]. Compounds, including prodigiosin, 2-methyl-3-propyl-prodiginine, 2-methyl-3-butyl-prodiginine, and 2-methyl-3-heptyl-prodiginine were used as antimicrobial agents against six pathogenic bacteria: *E. coli* ATCC 8739, *P. aeruginosa* 68 [42], Methicillin-resistant *S. aureus* ATCC6538, *S. paratyphi* B, *B. subtilis* CGMCC 1.88, and *C. albicans* ATCC 10,231. The test compounds were dissolved in DMSO with a final concentration of 5 mg/mL. For the agar diffusion assay, the indicating strains were grown in liquid medium overnight. The next morning, the cultures were mixed with 55 °C LB or YPD agar by 0.1% and poured on plastic plate. When the agar plates were solid, 5 µL (25 µg) of testing compounds were dipped. The LB agar plates were kept in the incubator at 37 °C for 16 h and the YPD agar plates were kept at 28 °C for 30 h. The MIC test was performed as below. The indicating strains were grown overnight. The next morning, the cell density (OD_625_) was recorded and diluted to 0.08–0.13. The diluted culture was inoculated in fresh medium by 1% and transferred to a 96-well plate with 100µL of culture in each well. The first well of each row on the 96-well plate contained test compounds with a concentration of 400 µg/mL. After diluting each well from the left to the right by two times, the 96-well plates were kept at 28 or 37 °C overnight. The next morning, the 96-well plates were collected and checked with the naked eye.

### 4.10. Fermentation of Prodigiosin

One single colony of *H. chejuensis* NBU794 was inoculated in 3 mL of 50% R2A liquid medium and grown at 28 °C at a speed of 180 rpm/min overnight. On the second day, the culture was used as the seed to inoculate by 1% into four different liquid mediums (50 mL), including standard ISP4, standard ISP4 with 1 g HP20 beads, modified ISP4 (starch is replaced with equal amount of glucose) with 1 g HP20 beads added, and modified ISP4 (starch is replaced with an equal amount of sucrose) with 1 g HP20 beads. The fermentation was performed at 28 °C with a speed of 180 rpm/min for three days. When the fermentation ceased, the culture without HP20 was extracted with ethyl acetate, evaporated, and suspended in MeOH for HPLC analysis. Regarding the culture with HP20 beads, the beads were filtered first, cleaned with water, and then washed with MeOH. The eluted extracts were evaporated and suspended in MeOH for HPLC analysis. The extract samples were injected into a Luna C18 column (250 × 4.6 mm, 5 μm, Phenomenex) in 10 μL batches. The column was eluted in a gradient from 30% aqueous MeCN to 65% MeCN in 18 min (LiChrosolv, Merck) at 25 °C and washed with 100% MeCN between the injections. The standard curve of prodigision was created based on different concentrations of prodigision and their corresponding peak integration areas from HPLC analysis at UV 530 nm (Appendix A).

## 5. Conclusions

Herein, the genome mining of the *Hahella* genus revealed that each genome contains up to 12 SMBGCs on average, which is much more than the average number (5) derived from the analysis of 200,000 microbial genomes, indicating that *Hahella* is a good resource for finding new natural products. The omics-based analysis uncovered a number of prodiginine derivatives, including two new compounds, 2-methyl-3-propyl-4-O-methyl-prodiginine and 2-methyl-3-pentyl-4-O-methyl-prodiginine, and chejuenolide A-C from a new isolate strain *H. chejuensis* NBU794, which enriched the chemical diversity in *Hahella*.

## Figures and Tables

**Figure 1 marinedrugs-20-00269-f001:**
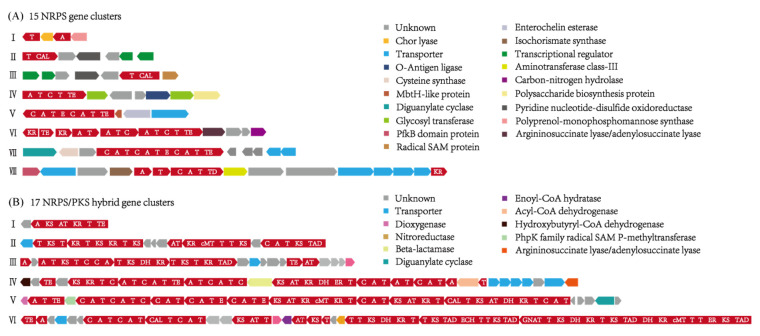
NRPS gene clusters and NRPS/PKS hybrid gene clusters are identified in *Hahella ganghwensis* DSM 17,046, *Hahella chejuensis* KCTC 2396, *Hahella chejuensis* KA22, *Hahella chejuensis* HN01, and *Hahella* sp.CCB-MM4. (**A**) A total of 15 NRPS gene clusters are identified and classified into eight groups based on the gene composition. (**B**) A total of 17 NRPS/PKS hybrid gene clusters are identified and classified into six groups based on the gene composition. The functional domains are indicated in bold letters: A, adenylation domain; AT, acyl-transferase; CAL, co-enzyme A ligase domain; C, condensation domain; cMT, carbon methyltransferase; DH, dehydratase; E, epimerization; ECH, enoyl-CoA hydratase/isomerase; ER, enoylreductase domain; GNAT, GCN5-related N-acetyltransferases domain; KR, ketoreductase; KS, ketosynthase; T, thiolation domain (peptidyl carrier protein); TAD, Trans-AT docking domain; TD, Terminal reductase domain; TE, thioesterase. The modular genes are marked with red color, while the other functional enzymes are indicated by squares with different colors.

**Figure 2 marinedrugs-20-00269-f002:**
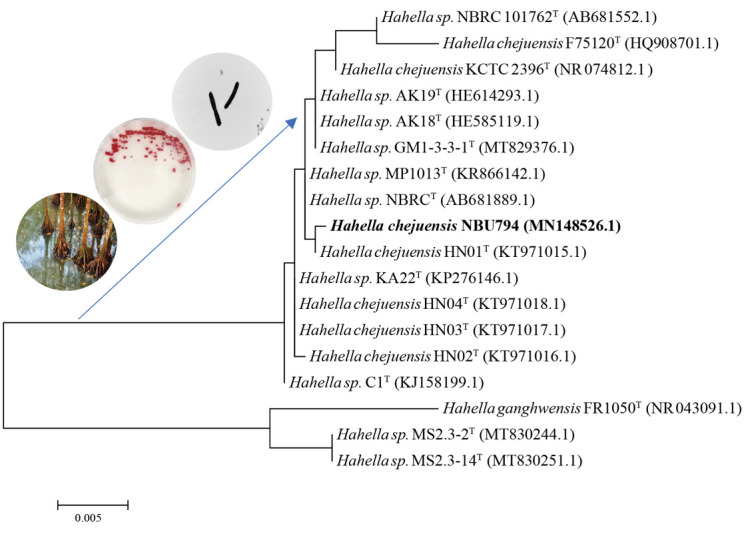
Neighbor-joining phylogenetic tree based on 16S rRNA gene sequences showing the positions of strain NBU794 and other strains in the genus *Hahella*. Bar, 0.005 represents nucleotide substitution rate (Knuc) units. Strains *H. chejuensis* NBU794 from this study are highlighted in bold. The figures of sample collection, strain morphology on the agar plate, and transmission electron microscopy results are listed on the left top corner.

**Figure 3 marinedrugs-20-00269-f003:**
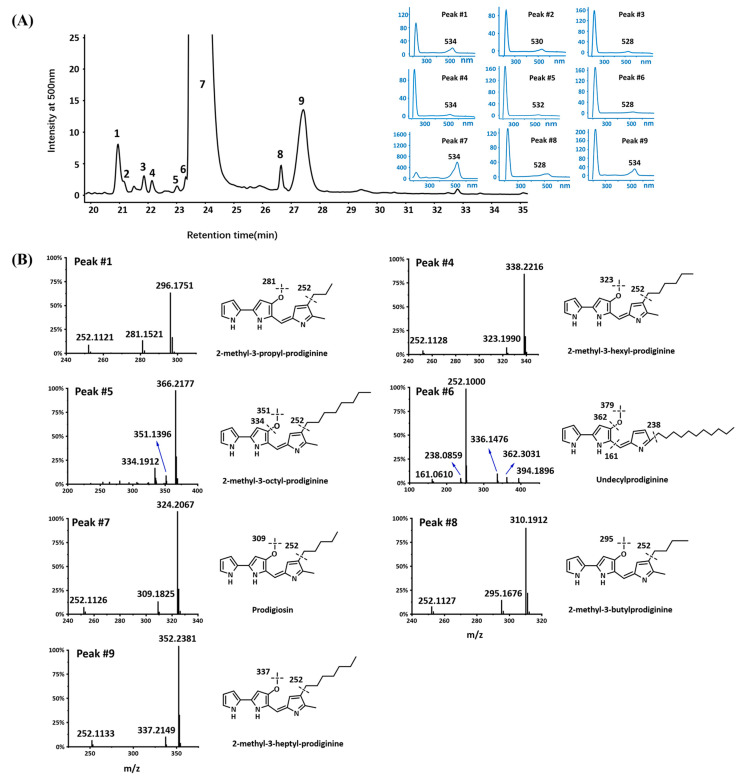
LC-MS/MS analysis of prodiginine derivatives in *H. chejuensis* NBU794. (**A**) The nine UV peaks corresponding prodiginine derivatives were marked with arabic numbers, and the UV−Vis spectra of nine eluent peaks are listed on the right. (**B**) The MS/MS spectrums are listed on the left and compound structures with fragment information are listed on the right. Peak #1: 2-methyl-3-propyl-prodiginine; Peak #4: 2-methyl-3-hexyl-prodiginine; Peak #5: 2-methyl-3-octyl-prodiginine; Peak #6: Undecylprodiginine; Peak #7: prodigiosin; Peak #8: 2-methyl-3-butyl-prodiginine; Peak #9: 2-methyl-3-heptyl-prodiginine.

**Figure 4 marinedrugs-20-00269-f004:**
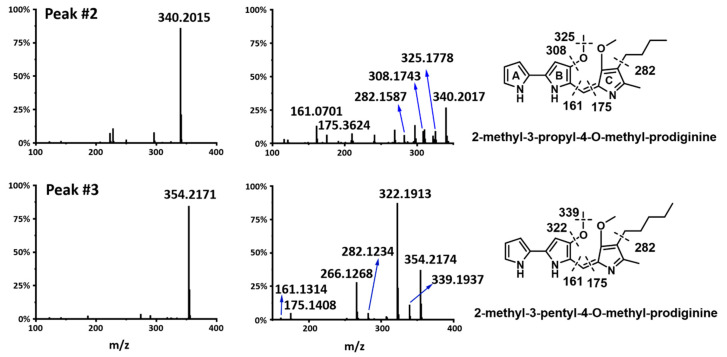
Structure elucidation of 2-methyl-3-pentyl-4-O-methyl-prodiginine, 2-methyl-3-octyl-prodiginine with LC-MS/MS. The product ion scans are listed on the left, the MS/MS spectrums are listed in the middle, and compound structures with fragment information are listed on the right.

**Figure 5 marinedrugs-20-00269-f005:**
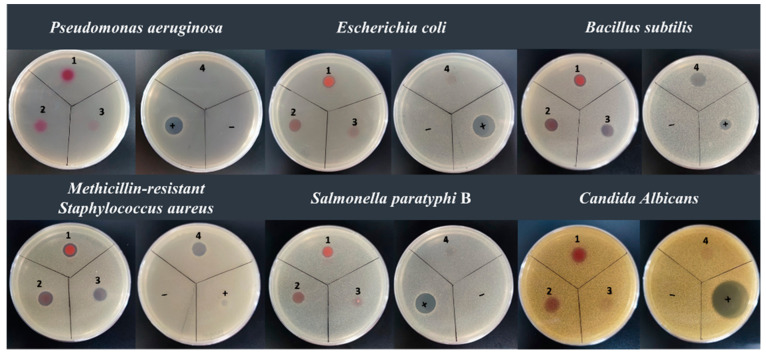
Antimicrobial assay of prodigiosin (1), 2-methyl-3-heptyl-prodiginine (2), 2-methyl-3-propyl-prodiginine (3), 2-methyl-3-butyl-prodiginine (4). (−): to use DMSO as a negative control. (+): to use nystatin as a positive control in the antifungal test against *Candida albicans*; to use polymyxin as a positive control in the antibacterial test against *Salmonella paratyphi* B, *Pseudomonas aeruginosa*, Methicillin-resistant *Staphylococcus aureus, Bacillus subtilis*, and *Escherichia coli*.

**Table 1 marinedrugs-20-00269-t001:** Minimum inhibitory concentration of prodigiosin and three congeners.

Compound	Minimum Inhibitory Concentration (µg/mL)				
	Methicillin-Resistant *Staphylococcus aureus*	*Bacillus subtilis*	*Escherichia coli*	*Salmonella paratyphi* B	*Candida albicans*
2-methyl-3-propyl-prodiginine	50	25	-	-	200
2-methyl-3-butyl-prodiginine	50	12.5	-	-	-
Prodigiosin	1.56	1.56	3.12	12.5	1.56
2-methyl-3-heptyl-prodiginine	1.56	6.25	12.5	25	6.25

## Data Availability

Not applicable.

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
