# Peer review of "Discovery of New Secondary Metabolites from Marine Bacteria Hahella Based on an Omics Strategy"

_marinedrugs, 2022, doi:10.3390/md20040269_

Round 1

Reviewer 1 Report

The manuscript by Shufen He et al. entitled: “Discovery of new secondary metabolites from marine bacteria Hahella based on omics-strategy” deals with a global approach to identify several new secondary metabolites in a new variety of a Hahella chejuensis bacterium.

The authors used a systematic secondary metabolites biosynthesis gene cluster bioinformatic research to identify 70 such gene clusters. About half of the genes in these gene clusters encode interesting non-ribosomal peptide synthases and polyketide synthases. This is a comprehensive study that brings significant amount of new data and that characterizes several new derivatives of the tripyrrolic compounds prodiginins and prodigiosin, und of the C17-ring chejuenolides. I have some specific comments and questions that I would like to see addressed in a revised version of the manuscript.

Specific comments

  1. None of the figures presented in the main text has a title and a legend. This omission is highly detrimental to the understanding of what each figure is aimed to illustrate? This should eb corrected in a revised version.
  2. Are genes encoding cytochromes P450 enzymes found in the newly discovered clusters?
  3. On page 3 line 115-116, what is exactly an epimerization domain in this context? Could the authors be more precise?
  4. Page 4 line 149. What is the source of this red color? Have the authors any idea on the corresponding dye?
  5. Page 5 line 180: what is the main difference in composition between all these culture media. Could the authors comment on that?
  6. Page 5 line 185-186. How can you be that affirmative in linking a mass of 324 dalton and the molecular formula you present? Are they not other chemicals with that same mass?
  7. Page 8 line 251. The 1.56 µg/mL and 50 µg/mL concentrations correspond to molarities of about 5 µM and 155 µM. These values are not that small and lower active concentrations (rather in the nM range) are most of the time necessary for a possible efficient effect in biological systems. Could the authors comment on that concentration gap?
  8. I do recommend the authors to place in their revised manuscript the chemical structures of the newly identified NBU794 compounds in the main text in a new figure, not as a supplemental figure. This is to me an important result that deserves a figure on its own.

Minor comments

  1. The English should be checked throughout the text. For instance, on page 1 line 45, the “were discovery” is not proper English, or a word is missing.
  2. The sentence page 3 lines 170-171 as such is not understandable. Please reformulate it.
  3. At several occurrences, the names of bacteria are not always written in italics. Please correct this.
  4. Page 3 line 133. There is a typo: “siderphore” should be: “siderophore”.
  5. Page 5 line 183. “We isolated this peak” is lab’s jargon not correct in English: a “peak” cannot be isolated but the chemical that results, when detected, in this peak. Please change this sentence accordingly and also to other similar occurrences in the text.
  6. Page 5 line 181. The word “huge” here is not appropriate, “prominent” would be better.
  7. Page 8 line 263. HP20 beads. Could the authors explicit what are HP20 beads for the non-specialist reader?
  8. Page 9 line 294. There is a typo here. “21” should be “21st”.
  9. The supplemental figure S4 is the same as figure 2. Why such a duplication of the same info?

Author Response

Dear Reviewer ,

Thank you very much for your time involved in reviewing the manuscript and your very encouraging comments on the merits.

To facilitate this discussion, we first retype your comments in italic font and then present our responses to the comments.

Specific comment 1

None of the figures presented in the main text has a title and a legend. This omission is highly detrimental to the understanding of what each figure is aimed to illustrate? This should eb corrected in a revised version.

 Response 1: Thanks for the reviewer’s comment. We feel sorry to make this mistake. The figure titles and legends have been added in the manuscript between line 608 and line 645.

Specific comment 2

Are genes encoding cytochromes P450 enzymes found in the newly discovered clusters?

Response 2: Thanks for the reviewer’s question. We go through all the gene clusters and only find one case in the PBDE type gene cluster from Hahella sp. CCB-MM4. The searching result has been attached as follow.

Specific comment 3

On page 3 line 115-116, what is exactly an epimerization domain in this context? Could the authors be more precise?

Response 3: Thanks for the reviewer’s comment. We rephrased the sentence and marked with the red color between line 108 and line 111.

Specific comment 4

Page 4 line 149. What is the source of this red color? Have the authors any idea on the corresponding dye?

 Response 4: Thanks for the reviewer’s question. The red color actually results from the Prodiginine derivatives, including the well-known compound prodigiosin. We isolated prodigiosin (the major component in the fermentation culture) and resuspend it in MeOH. The solution was red between pH1 and pH7, and showed pink color at pH10 and yellow at pH12. The figure is attached as follow.

Specific comment 5

Page 5 line 180: what is the main difference in composition between all these culture media. Could the authors comment on that?

Response 5: Thanks for the reviewer’s question. M9 medium is one kind of minimal medium. ISP4 is one kind of medium that is rich in carbon source, starch. R2A is one common-used medium for marine bacteria, and it contain a rich nitrogen source. 2216E is also widely used for growing marine bacteria. Except for the nitrogen source, 2216E contains a variety of metal ions. All four mediums contain different nutrient composition and metal ions. We applied them in order to induce the expression of secondary metabolite biosynthetic gene clusters in Hahella sp. NBU794 under different culture conditions.

Specific comment 6

Page 5 line 185-186. How can you be that affirmative in linking a mass of 324 dalton and the molecular formula you present? Are they not other chemicals with that same mass?

 Response 6: Thanks for the reviewer’s comment. In fact, we confirmed the compound with a molecular weight of 324 as prodigiosin based two factors. First, we compared it with prodigiosin standard in the Hahella chenjuensis KCTC 2396, and both have the same retention time and UV-Vis spectrum. Second, the error between HRMS data of compound (324.2067) and prodigiosin molecular mass (324.2070, [M+H]+) is less than 5ppm.

Specific comment 7

Page 8 line 251. The 1.56 µg/mL and 50 µg/mL concentrations correspond to molarities of about 5 µM and 155 µM. These values are not that small and lower active concentrations (rather in the nM range) are most of the time necessary for a possible efficient effect in biological systems. Could the authors comment on that concentration gap?

Response 7: Thanks for the reviewer’s comment. We agree with the reviewer that the tested IC50 of Prodigiosin and its congeners is not that promising compared with other well-known antibiotics. When we performed the anti-microbial test, the light was hard to be excluded. Prodigiosin and its congeners are very sensitive to light, we think this might result in a higher IC50. We believe their IC50 might be lower than tested data in this study. Besides, we think the IC50 value(5µM) is not that bad and even better than many naturally occurring antibiotics. If considering the light sensitivity and good anti-microbial activity together, we believe Prodigiosin and its congeners have a potential application in food industry and environmental issues (water bloom), because they not only own good anti-microbial activity, but also cause no antibiotics residues through the natural degradation.

Specific comment 8

I do recommend the authors to place in their revised manuscript the chemical structures of the newly identified NBU794 compounds in the main text in a new figure, not as a supplemental figure. This is to me an important result that deserves a figure on its own.

 Response 7: Thanks for the reviewer’s comment. We made a new figure to exhibit the structures of new compounds, which is Figure 4.

Minor comment 1

The English should be checked throughout the text. For instance, on page 1 line 45, the “were discovery” is not proper English, or a word is missing.

Response 1: Thanks for the reviewer’s comment. We go through the manuscript and make some necessary modification on the English. The error on page 1 line 45 was solved and new sentence was marked with red color on line 51.

Minor comment 2

The sentence page 3 lines 170-171 as such is not understandable. Please reformulate it.

 Response 2: Thanks for the reviewer’s comment. We rewrite the sentence between line 154 and line 157 and mark them with red color.

Minor comment 3

At several occurrences, the names of bacteria are not always written in italics. Please correct this.

Response 3: Thanks for the reviewer’s comment. We have changed the spelling errors in the manuscripts and mark them with red color.

Minor comment 4

Page 3 line 133. There is a typo: “siderphore” should be: “siderophore”.

 Response 4: Thanks for the reviewer’s comment. We have changed the word on line 121 and mark it with red color.

Minor comment 5

Page 5 line 183. “We isolated this peak” is lab’s jargon not correct in English: a “peak” cannot be isolated but the chemical that results, when detected, in this peak. Please change this sentence accordingly and also to other similar occurrences in the text.

 Response 5: Thanks for the reviewer’s comment. We change the errors according to review suggestion and mark the changes with red color.

Minor comment 6

Page 5 line 181. The word “huge” here is not appropriate, “prominent” would be better.

 Response 6: Thanks for the reviewer’s comment. We change the “huge” into “prominent” on line 163.

Minor comment 7

Page 8 line 263. HP20 beads. Could the authors explicit what are HP20 beads for the non-specialist reader?

Response 7: Thanks for the reviewer’s suggestion. We add the explanation on HP20 on line 239.

Minor comment 8

Page 9 line 294. There is a typo here. “21” should be “21st”.

Response 8: Thanks for the reviewer’s comment. We change the error on line 265.

Minor comment 9

The supplemental figure S4 is the same as figure 2. Why such a duplication of the same info?

 Response 9: Thanks for the reviewer’s comment. Sorry to make the confusion, which results form the missing of Figure 2 legend. Actually, the trees in Figure 2 and Figure S4 are build up with different models (neighbor-joining and maximum likelihood) in order to make sure the predication on taxonomic status of Hahella sp. NBU794 is more reliable.

Reviewer 2 Report

In the current study, the authors identified several metabolites from the newly identified species H. chejuensis NBU794 and evaluated the anti-bacterial activities of the isolates. The results were of some interest. Some major issues should be addressed before further consideration.

  1. The peaks 2 and 3 were identified as new compounds. And these compounds were purified with preparative HPLC. The NMR and other spectral data (IR, UV, et al) should be included.
  2. In the anti-bacterial assay, no positive control was evaluated.
  3. The manuscript was poorly written. A thorough editing by a native English speaker must be carried out.
  4. The reference part was missing.
  5. There was lack of figure legend.
  6. the email of all the authors should be supplied.

Author Response

Dear Reviewer ,

Thank you very much for your time involved in reviewing the manuscript and your very encouraging comments on the merits.

To facilitate this discussion, we first retype your comments in italic font and then present our responses to the comments.

Comment 1:

The peaks 2 and 3 were identified as new compounds. And these compounds were purified with preparative HPLC. The NMR and other spectral data (IR, UV, et al) should be included.

Response 1: Thanks for the reviewer’s comment. We agree with the reviewer that the NMR and other spectral data are very important to solidate the compound structure. We have tried many times to purify those two new compounds with preparative HPLC, but failed. They are minor components in the crude extracts and stack with other compounds, so it is not that easy to purify enough amount for NMR. At the same time, two new compounds are very sensitive to light. Even though we tried to avoid the light during the purification and lyophilization, the NMR data still showed a mixture. Therefore, we introduce LC-MS/MS to identify the new structure, which is commonly used to identify the structure of Prodigiosin and its congeners.

Comment 2:

In the anti-bacterial assay, no positive control was evaluated.

Response 2: Thanks for the reviewer’s question. Actually, we evaluated the positive control in the anti-bacterial assay. We feel sorry to cause this confusion by missing the legend from Figure 5. In the revised manuscript, we added the missing figure legend. In the anti-microbial test, such as the agar diffusion test, we added polymyxin and nystatin as the positive control in the anti-bacteria and antifungal test, respectively.

Comment 3:

The manuscript was poorly written. A thorough editing by a native English speaker must be carried out.

Response 3: Thanks for the reviewer’s comment. We go through the manuscript and make a plenty of modification on the English writing.

Comment 4:

The reference part was missing.

Response 4: Thanks for the reviewer’s comment. We feel very sorry about this mistake. The reference has been added between line 517 and line 607.

Comment 5:

There was lack of figure legend.

Response 5: Thanks for the reviewer’s comment. We feel very sorry about this mistake. The Figures with legends have been added between line 606 and line 643.

Comment 6:

the email of all the authors should be supplied.

Response 6: Thanks for the reviewer’s comment. We have added the email address of all the authors.

Round 2

Reviewer 2 Report

All my concerning has been addressed.